# Preliminary Characterization of Triatomine Bug Blood Meals on the Island of Trinidad Reveals Opportunistic Feeding Behavior on Both Human and Animal Hosts

**DOI:** 10.3390/tropicalmed5040166

**Published:** 2020-11-04

**Authors:** Alexandra Hylton, Daniel M. Fitzpatrick, Rod Suepaul, Andrew P. Dobson, Roxanne A. Charles, Jennifer K. Peterson

**Affiliations:** 1Department of Ecology and Evolutionary Biology, Princeton University, Princeton, NJ 08544, USA; aeeakes@gmail.com (A.H.); dobson@princeton.edu (A.P.D.); 2Department of Pathobiology, School of Veterinary Medicine, St. George’s University, True Blue, Grenada; dfitzpat@sgu.edu; 3College of Medicine, University of Oklahoma, Oklahoma City, OK 73117, USA; 4Department of Basic Veterinary Sciences, School of Veterinary Medicine, Faculty of Medical Sciences, The University of the West Indies, Trinidad and Tobago, West Indies; Rod.Suepaul@sta.uwi.edu (R.S.); Roxanne.Charles@sta.uwi.edu (R.A.C.); 5University Honors College, Portland State University, Portland, OR 97207-075, USA

**Keywords:** chagas disease, *Trypanosoma cruzi*, triatomine bugs, *Panstrongylus geniculatus*, *Rhodnius pictipes*, Trinidad and Tobago, West Indies, vector host-feeding preferences, blood meal analysis

## Abstract

Chagas disease is a neglected tropical disease caused by infection with *Trypanosoma cruzi.* The parasite is endemic to the Americas, including the Caribbean, where it is vectored by triatomine bugs. Although Chagas disease is not considered a public health concern in the Caribbean islands, studies in Trinidad have found *T. cruzi*-seropositive humans and *T. cruzi*-infected triatomine bugs. However, little is known about triatomine bug host preferences in Trinidad, making it difficult to evaluate local risk of vector-borne *T. cruzi* transmission to humans. To investigate this question, we collected triatomine bugs in Trinidad and diagnosed each one for *T. cruzi* infection (microscopy and PCR). We then carried out a blood meal analysis using DNA extracted from each bug (PCR and sequencing). Fifty-five adult bugs (54 *Panstrongylus geniculatus* and one *Rhodnius pictipes*) were collected from five of 21 sample sites. All successful collection sites were residential. Forty-six out of the 55 bugs (83.6%) were infected with *T. cruzi*. Fifty-three blood meal hosts were successfully analyzed (one per bug), which consisted of wild birds (7% of all blood meals), wild mammals (17%), chickens (19%), and humans (57%). Of the 30 bugs with human blood meals, 26 (87%) were from bugs infected with *T. cruzi*. Although preliminary, our results align with previous work in which *P. geniculatus* in Trinidad had high levels of *T. cruzi* infection. Furthermore, our findings suggest that *P. geniculatus* moves between human and animal environments in Trinidad, feeding opportunistically on a wide range of species. Our findings highlight a critical need for further studies of Chagas disease in Trinidad in order to estimate the public health risk and implement necessary preventative and control measures.

## 1. Introduction

Chagas disease (American trypanosomiasis) is a neglected tropical disease (NTD) caused by infection with the protozoan parasite *Trypanosoma cruzi*. Symptoms of the disease include cardiac, gastrointestinal, and/or neurological alterations. An estimated six million people are currently infected with *T. cruzi*, with 25 million more at risk [1].

*T. cruzi* is endemic to the Americas, where it cycles between mammalian hosts and triatomine bug vectors. Vector-borne *T. cruzi* transmission to humans can occur via contact with infected triatomine excrement during or after the bug takes a blood meal, or orally, through ingestion of *T. cruzi*-contaminated food or beverages [2,3,4,5]. Food or beverage contamination with *T. cruzi* can occur in a number of ways, but most commonly an infected bug either falls into or defecates on the food. Other human *T. cruzi* infection routes include contaminated blood transfusions or organ transplants, transplacental transmission, and laboratory accidents [1].

Triatomine bugs acquire *T. cruzi* infection by taking blood meals from infected mammals in sylvatic or domestic/peridomestic transmission cycles. These transmission cycles can be connected, especially in disturbed or fragmented habitats [6], by triatomines that forage opportunistically in both their sylvatic habitats and nearby residential areas, including human homes [7,8]. Due to the existence of distinct *T. cruzi* transmission cycles, the presence of triatomine bugs in a region does not necessarily mean that the region is Chagas-endemic. Such is the case in the Caribbean islands, where triatomine bugs are widely distributed [9], but believed to be primarily sylvatic, making infrequent contact with humans [9,10,11]. Although *T. cruzi*-positive serology in humans has been reported in at least three Caribbean islands (Curacao, Jamaica, and Trinidad [9,12,13,14]), there have been few diagnosed clinical cases. As such, Chagas disease is not recognized by major public or global health organizations as endemic to any Caribbean island [10,15].

Of particular interest for Chagas disease in the Caribbean is the island of Trinidad, of the dual-island Republic of Trinidad and Tobago. Trinidad has geological origins on the continental shelf of South America [16], and shares much of its flora and fauna with Chagas-endemic, neighboring Venezuela (located just eight miles west of Trinidad, Figure 1). Included in the fauna found in both Trinidad and Venezuela are six triatomine bug species: *Eratyrus mucronatus*, *Microtriatoma trinidadensis*, *Rhodnius pictipes*, *Panstrongylus geniculatus*, *Panstrongylus rufotuberculatus*, and *Triatoma rubrofasciata* [17,18,19]. All species are competent vectors of *T. cruzi* [11], but *P. geniculatus* is believed to be the most abundant [17], and of the highest epidemiological importance [3,20,21,22,23,24].

Although there are only a few studies of Chagas disease in Trinidad, they provide compelling evidence that *T. cruzi* may not be strictly enzootic on the island. Cardiologist Boris Fistein first suspected that Chagas disease was endemic to the island in the early 1960s, when he observed the clinical picture of the disease in several patients with congestive heart failure not attributable to common causes [12,13]. Fistein found that some of these patients were seropositive for *T. cruzi* [12,13,25], and a follow-up study of the vector yielded 79 triatomine bugs collected in and around the patients’ homes. Of 69 bugs examined, 35 (50.72%) were infected with *T. cruzi* [25]. A later study of sera collected from venereal disease clinics and antenatal clinics (meant to represent the general population) found a *T. cruzi* seroprevalence of 0.45% [12]. Several years later, a larger study of triatomines in Trinidad yielded the same species infected with *T. cruzi* distributed widely throughout the island [17]. Finally, in a 1992 serological survey of 192 cardiac patients in south Trinidad, 72 (37.50%) tested positive for *T. cruzi* antibodies [26]. The authors report that 49 of the 72 patients (68.06%) had *T. cruzi* trypomastigotes in peripheral blood, although this number seems unusually high.

While these findings provide mounting evidence of a vector-borne Chagas disease transmission cycle existing in Trinidad, there are no known published data demonstrating the species on which *T. cruzi*-infected triatomine bugs are feeding, other than a single wild rat found infected with *T. cruzi* in 1963 [27]. Identification of the host species with which triatomines come into contact by taking a blood meal can give us a sense of the frequency with which the bugs invade domestic environments, and in turn, the risk of vector-borne Chagas disease. For example, a diet of primarily wild animal hosts would suggest that there is little risk of the disease on the island, while a diet involving domestic animals and/or humans would imply that Chagas disease may be of local epidemiologic importance. As such, we asked, from which species are *T. cruzi*-infected triatomines in Trinidad taking blood meals? We aimed to use our findings to further characterize the current *T. cruzi* transmission cycle in Trinidad, and in turn, better understand the risk of Chagas disease transmission on the island.

## 2. Materials and Methods 

### 2.1. Overview

Triatomine bugs were sampled on the island of Trinidad between May and August, 2016. Bugs were collected using mouse-baited traps placed near animal resting/nesting sites and artificial light sources (Figure 2), in addition to manual collection when a bug was found resting on a wall or other surface. Public submissions were also accepted. Upon collection, bugs were examined for *T. cruzi* infection before undergoing blood meal analysis to identify a species on which each bug had recently taken a blood meal.

### 2.2. Animal Use Ethics Statement

The mouse-baited trap protocol used in this study was reviewed and approved by The Institutional Animal Care and Use Committee (IACUC) of Princeton University (protocol # 2065F-16) and St. George’s University (approval number 16010-R), and the Campus Ethics Committee at the University of the West Indies (reference number CEC172/04/16).

### 2.3. Study Sites 

A total of 21 sites were sampled across 11 villages concentrated mostly in northern and central Trinidad (site-specific details in Appendix A). All sites except one were residential or very close to human disturbance (i.e., residential, farm, logging, alongside a road, etc.). Within each site, particular attention was paid to *Attalea maripa* palms known locally as ‘cocorite,’ which are known habitat for local fauna. We sampled for one night per location, with repeat sampling of a site occurring after a successful collection.

### 2.4. Triatomine Bug Collection 

We used three methods to collect triatomine bugs: mouse-baited traps, manual trapping, and community collection. Mouse-baited traps (Figure 2, [28]) were placed in animal resting sites, as triatomines tend to live near their food source. These sites included rat nests, opossum nests, bat-infested structures, and 18 *A. maripa* palms (details for each site in Appendix A). In residential areas, we placed traps near light sources in building exteriors. 

Mouse-baited traps consisted of a ‘T-shaped’ PVC pipe (3” diameter) covered in Duck^®^ Brand Indoor Heavy Traffic Carpet Tape, which is a double-sided adhesive material (Figure 2). A Swiss albino mouse was placed inside the trap just before use, along with shredded paper for bedding and a piece of potato or fruit that served as a food and water source. Trap openings were closed by a two-inch section of PVC pipe covered in chicken wire and fine mesh. Traps were placed at dusk (5:00 p.m.–8:00 p.m.) and retrieved the next day in the morning (7:00 a.m.–9:00 a.m.). After each use, the mouse was removed from the trap and returned to its cage. Traps were cleaned and disinfected, and the water source, bedding, and tape were changed before reusing the trap. Each mouse was used in a trap only once per week. Mice were obtained from the laboratory animal rearing facility of the University of the West Indies Veterinary School.

In addition to trapping, we collected bugs opportunistically when found resting on exterior walls near artificial lights. We also accepted public submissions, as described in [29]. Chagas disease information pamphlets with pictures of local bug species were distributed (Appendix A) to those residing near positive collection sites.

### 2.5. Specimen Processing and Molecular Analyses

All protocols were performed under carefully controlled laboratory conditions. DNA extraction and blood meal analyses were carried out in separate laboratories to minimize cross-contamination risk. Bugs were processed in the Pathology and Parasitology Laboratory at the School of Veterinary Medicine of the University of the West Indies (UWI), St. Augustine, Trinidad. Triatomine species were determined using taxonomic keys and images from Lent and Wygodzinsky [30], Carcavallo et al. [31], and Patterson et al. [32]. *T. cruzi* infection in live bugs was diagnosed through direct microscopy observation (DMO) and staining of bug excrement and stomach contents (Figure 3), while *T. cruzi* infection in dead bugs was diagnosed using PCR. For DMO, 10 µL of Phosphate Buffer Solution (PBS) was mixed with 10 µL of excrement from each bug. A 5 µL aliquot of the solution was examined microscopically at 40× magnification. Trypanosomes were identified by their movement and morphology. Bugs were stored in 70% ethanol (EtOH) for approximately one to three days, until molecular analysis was carried out.

DNA was isolated from each bug using the Qiagen DNeasy^®^ Blood and Tissue DNA isolation kit, using a modified protocol for tissue extraction (described in [33]). PCR diagnosis of *T. cruzi* was performed for all bugs using the TcZF/R primer (5′-GCTCTTGCCCACAAGGGTGC-3′ and 5′-CCAAGCAGCGGATAGTTCAGG-3′, [34]), which targets a 182 bp sequence of *T. cruzi* satellite DNA. Following optimization, the final recipe consisted of 1.25 µL of each TcZ-F/R, 1 µL DNA, 21.5 µL dH_2_O, and one GE Healthcare illustra PuReTaq Ready-To-Go PCR bead. PCR conditions were as follows: 94 °C for 5 min, followed by 40 cycles of 94 °C for 20 s, 57 °C for 10 s, 72 °C for 30 s, with a final extension of 72 °C for 7 min, and held at 4 °C. All samples were visualized in a 1.5% agarose gel to confirm amplicon size. To confirm that the amplified PCR product was indeed *T. cruzi*, we extracted three of the 182 bp bands from the agarose gel following PCR using the Qiagen QIAEX II Gel Extraction Kit, and sent the bands for Sanger sequencing at the Plant-Microbe Genomics Facility at The Ohio State University. We edited sequences with Chromas Lite 2.6.4, and compared them to known sequences in NIH-NCBI GenBank using the Basic Local Alignment Search Tool (BLAST). These analyses confirmed the DNA sequence amplified was from *T. cruzi*. 

### 2.6. Blood Meal Analyses 

Blood meal analyses were performed in the Molecular Biology Lab at the St. George’s University School of Veterinary Medicine in True Blue, Grenada. We amplified a 360−380 bp fragment of the vertebrate cytochrome b (cytb) gene from the DNA of each bug using the L14841 primer (5’-CCCCTCAGAATGATATTTGTCCTCA-3′) and the H15149 primer (5′-CCATCCAACATCTCAGCATGATGAAA-3′) [35]. These primers are vertebrate-specific, and are used frequently to preferentially amplify host DNA from blood meals in hematophagous insects [35]. After optimization, the final recipe used was: 1.25 µL of each primer, 2 µL of MgCl_2_, 3 µL of bug DNA, 17.5 µL of dH_2_O, and one GE Healthcare illustra PuReTaq Ready-To-Go PCR bead. All PCRs included appropriate controls. Each sample was run under the following thermal cycling conditions: preheat at 95 °C for 5 min, then 50 cycles of 95 °C for 30 s, 50 °C for 45 s, 72 °C for 90 s, followed by 72 °C for 7 min, then hold at 4 °C. All samples were electrophoresed, extracted from gels, sequenced, edited, and compared to known GenBank entries, as described above. The cut off for accepting an identity match in BLAST was ≥90%, and an e-value below 1e−10. (The e-value describes the number of hits one can expect by chance, given database size.) In instances where more than one species had an identity match in BLAST over >90%, the GenBank sequence entry with highest e-value was called. Overlapping, short, or messy sequences were re-run using the same blood meal analysis protocol but with avian-specific and mammal-specific primers that amplified a 508 and 772 bp fragment of the cytb gene, respectively [36]. Sequence data are found in Appendix A.

## 3. Results

A total of 55 adult triatomine bugs were collected (54 *Panstrongylus geniculatus* and one *Rhodnius pictipes*) in five of the 11 villages sampled (Table 1). All successful collection spots were in or around human dwellings. 

No bugs were collected in traps placed in or near animal nesting sites. The majority of bugs (46 of 55; 83.6%) were collected from a single rural area with scattered human dwellings located adjacent to secondary forest. All triatomines collected were in the adult stage; 18 were female and 37 were male.

### 3.1. T. cruzi Infection 

*T. cruzi* DNA was amplified from 46 out of the 55 bugs collected (83.6%). Of these bugs, six were also diagnosed through microscopy (Figure 3; 12 bugs in total were alive upon arrival to the lab). Bugs diagnosed microscopically (in addition to PCR diagnosis) were from Coal Mine (N = 4), Matura (N = 1), and Mt. Harris (N = 1). All successful collection villages (N = 5, Table 1) yielded at least one bug infected with *T. cruzi* (Table 1 and Table 2).

### 3.2. Blood Meal Analyses 

We identified a recent blood meal host species for 53 of the 55 bugs (96.4%; Figure 4, Table 2). Thirty bugs (56.6%) fed on humans, of which 26 (86.7%) were infected with *T. cruzi*, including the single *R. pictipes* individual. The second most common blood meal hosts were chickens (10 bugs; 18.9%), followed by red howler monkeys (2 bugs, 4.0%), red-rumped agoutis (2 bugs, 4.0%), and blue-headed parrots (2 bugs, 4.0%). One blood meal each was identified from a porcupine, mongoose, opossum, water rat, spiny rat, woodpecker, and owl (detailed in Table 2). Overall, nine bugs (17.0%) fed on wild mammals and four bugs (7.5%) fed on wild birds. Each site yielded at least one bug with a human blood meal, and four of the five sites had at least one *T. cruzi*-infected bug with a human blood meal.

## 4. Discussion

In this study, we aimed to better understand the potential for vector-borne Chagas disease in Trinidad by characterizing the feeding preferences of *T. cruzi* vectors on the island. We found that at least one triatomine bug species on the island, *Panstrongylus geniculatus*, feeds on humans, chickens, and wild animals. While previous studies have found *T. cruzi* infection in humans [12,26] and triatomine bugs [17,25] in Trinidad, our study is the first to demonstrate that the bug feeds opportunistically across a wide range of host species, including humans, on the island.

### 4.1. P. geniculatus Breeding Habitat Remains Cryptic

All bugs in our study were adults collected from residential areas, despite efforts to sample a wide range of animal resting sites where the bugs might breed (i.e., cocorite palms, opossum nests, bat colonies, etc., Appendix A). Still, approximately one-quarter of identified blood meals came from wild animals. Prior studies of *P. geniculatus* in Trinidad found nymphs only in caves and burrows of the nine-banded armadillo (*Dasypus novemcinctus*) [12,17], while in Venezuela, *P. geniculatus* was also associated with the common or black-eared opossum (*Didelphis marsupialis*) [37]. While our blood meal analyses did not reveal either of these host species, we did identify blood meals from several other nesting animals in addition to arboreal species, both of which could provide suitable habitats for the bugs. Thus, it is possible that *P. geniculatus* in Trinidad select breeding sites in a similar manner to how they feed—opportunistically, across a range of species. 

Our failure to find juvenile triatomine bugs in residential areas suggests that the bugs forage, but do not breed, in domestic settings. Nonetheless, it important to note that almost one-fifth of identified blood meals in this study were from chickens in residential settings. Although birds are not competent *T. cruzi* hosts, they are a common food source for triatomines, which may in turn amplify *T. cruzi* transmission by increasing carrying capacity of vector populations [38]. Moreover, chicken coops are often found infested with triatomine bugs in other regions, and have even been used as sentinels for Chagas disease [39]. Thus, it will be of critical importance to sample more widely across residential settings in Trinidad in order to understand the potential for triatomines to breed in domestic settings, especially those with chickens. 

### 4.2. Are Wildlife Population Declines Driving P. geniculatus Host Shifts?

Although we did not find blood meals from nine-banded armadillos in this study, carcasses with *T. cruzi* amastigotes are occasionally brought by hunters to the Veterinary Pathology lab at the University of the West Indies. A larger study would likely yield blood meals from armadillos and possibly *Didelphis* opossums as well, although it is unclear if either species would carry the same importance for *P. geniculatus* as they did 40 years ago (when triatomines were last studied in Trinidad, [17]). Nine-banded armadillos and *Didelphis* opossums are regularly hunted in Trinidad as bushmeat, for which there is high demand; an opossum carcass in Trinidad currently sells for TT$ 300−500 (USD $40−$70). Thus, it is possible that these species are now less abundant, which could be driving *P. geniculatus* into new areas to nest and forage, a phenomenon observed in *P. geniculatus* in the Amazon basin [40,41]. Further studies of wildlife population decline in Trinidad would help identify if reduced wild host species abundance is causing the bugs to leave their nesting sites to feed on humans and domestic animals. 

### 4.3. P. geniculatus: A Cosmopolitan Species of Emerging Importance 

*P. geniculatus* is highly adaptive in both its feeding and habitat preferences. The species is widely distributed in Latin America across a multitude of landscapes [32]. Though historically associated with enzootic *T. cruzi* transmission, the species is increasingly reported in human homes and with human blood meals [21,22,23]. While once considered to be incapable of domiciliation, the species is now considered to be domiciliated or in the process of domiciliation in several regions [20,21,42], which is reviewed in [43]. Additionally, *P. geniculatus* is no longer limited to forested areas; in the Metropolitan district of Caracas, Venezuela, thousands of *P. geniculatus* were found in both poor and wealthy areas of 31 out of 32 parishes, with nymphs found in 14 of the parishes. The species has also been reported in urban centers in Brazil (Sao Paulo [44] and Rio de Janeiro [45]) and Bolivia (Cochabamba [24]).

### 4.4. Multiple Paths to T. cruzi Exposure in Trinidad

In addition, it is important to highlight that several different *T. cruzi* transmission routes could pose a risk to humans in Trinidad. The presence of *P. geniculatus* in areas of human use presents a risk for oral transmission of *T. cruzi*. Between 2007 and 2009, there were three oral Chagas disease outbreaks at schools in Venezuela [46,47]. Subsequent analysis implicated *P. geniculatus* contamination of food or drinks in all three outbreaks [23,46,47]. Oral contamination does not require insect domiciliation or colonization of a residence; rather, it can occur merely by triatomines attracted to areas of food preparation by artificial lights. Handling of *T. cruzi*-infected wild animal carcasses by both hunters and researchers in Trinidad presents a *T. cruzi* transmission risk as well [48]. Transmission may also be possible by consuming undercooked, *T. cruzi*-infected bushmeat, although this is yet to be confirmed [49]. Considering the popularity of bushmeat in Trinidad, these transmission routes merit further investigation. Taken together, the multiple possible means of *T. cruzi* exposure in Trinidad suggest that *P. geniculatus* should be taken seriously on the island as a vector species of emerging epidemiologic importance.

### 4.5. Chagas Disease in Trinidad: Official Endemicity Would Bring Critical Resources

In our experience with community collection, we observed that community members wanted to know more about Chagas disease and its vectors. We were unable to find a local resource to reference, so we created the informational pamphlet (Appendix A). However, a pamphlet cannot compensate for the resources that are missing due to the country not being officially designated as ‘Chagas endemic’ by major global health organizations [10,15,50]. With official Chagas disease endemic status comes numerous resources [50], such as donated medications [51], blood bank monitoring support data coordination [52], and a framework for increasing *T. cruzi* screening in pregnant women [53,54]. Furthermore, endemicity provides the opportunity to participate in subregional intergovernmental Chagas disease control initiatives such as the Southern Cone Initiative (INCOSUR, [55]) and the Initiative for Chagas Disease Control in Central America and Mexico (IPCAM, [56]). Countries in these groups set joint regional goals, and annual meetings to share progress and challenges serve as a source of motivation and accountability. All of the aforementioned resources are available to ‘endemic’ countries with *T. cruzi* seroprevalences that are likely comparable to Trinidad, such as Costa Rica and Belize, and would be useful in Trinidad, where heart disease was the leading cause of death in 2015 [57]. 

### 4.6. Study Limitations

Our results are preliminary, with a small sample size that was limited in geographic distribution and duration. We hope to conduct a larger study in order to define the scope of *T. cruzi* transmission between humans and animals on the island, and determine whether our results are representative of the island as a whole. Additionally, seasonality may have contributed to our failure to find bugs in animal resting sites. Due to logistical constraints, the field work portion of this study took place during the rainy season, when palms, burrows, or other animal nests can be vacated due to humidity or inundation. Indeed, Fistein [12] found that the number of triatomines found inside houses was inversely proportional to both relative humidity and monthly rainfall. Accordingly, follow-up studies should be carried out in the rainy and dry seasons to accurately characterize *P. geniculatus* breeding sites in Trinidad. 

Finally, it must be kept in mind that a ‘bite’ (i.e., blood meal) from a *T. cruzi*-infected triatomine does not guarantee *T. cruzi* transmission, since the parasite is transmitted in vector excrement and not saliva. Thus, our finding of human blood meals in *T. cruzi*-infected bugs does not mean that the bugs transmitted *T. cruzi* to those humans. There is one published study of *P. geniculatus* defecation timing, and the authors found that the bugs do, on average, defecate while feeding [42], which is a key characteristic of epidemiologically important Chagas disease vector species. It is also important to mention that oral transmission is a risk simply with vector presence, regardless of whether or not the bug feeds or defecates on a human host. 

## 5. Conclusions and Future Directions

Although our findings are preliminary, our study is an important first step in furthering our understanding of *T. cruzi* dynamics in Trinidad. The results of our blood meal analysis suggest that *T. cruzi* in Trinidad may not be strictly enzootic, but rather could be a parasite of humans, domestic animals, and wild animals. Our findings support those of prior studies suggesting that Chagas disease in Trinidad needs to be more comprehensively studied, and perhaps re-considered in terms of its public health importance on the island. 

Future studies would benefit from taking a One Health approach entailing multidisciplinary research and outreach work focused on interactions between humans, domestic and wild animal populations, triatomine bugs, and the environment. Open questions include triatomine bug seasonality, regional variation, breeding habitats, and juvenile *T. cruzi* infection rates. Identifying mammalian hosts serving as *T. cruzi* infection sources for the bugs and measuring *T. cruzi* infection prevalence in wild and domestic animals will form a more complete picture of the *T. cruzi* transmission cycle on the island. Connections between human land use and *T. cruzi* transmission must be investigated. Epidemiologic surveys are needed to estimate baseline *T. cruzi* prevalence in humans in Trinidad, and it will be of critical importance to determine whether *T. cruzi* infection contributes to the high rates of heart disease on the island. If after further study, it is confirmed that Chagas disease is indeed a public health concern in Trinidad, a holistic prevention approach that integrates an awareness of animal hosts, human hosts, and the environment may be necessary in order to effectively address all aspects of this zoonotic NTD. 

## Figures and Tables

**Figure 1 tropicalmed-05-00166-f001:**
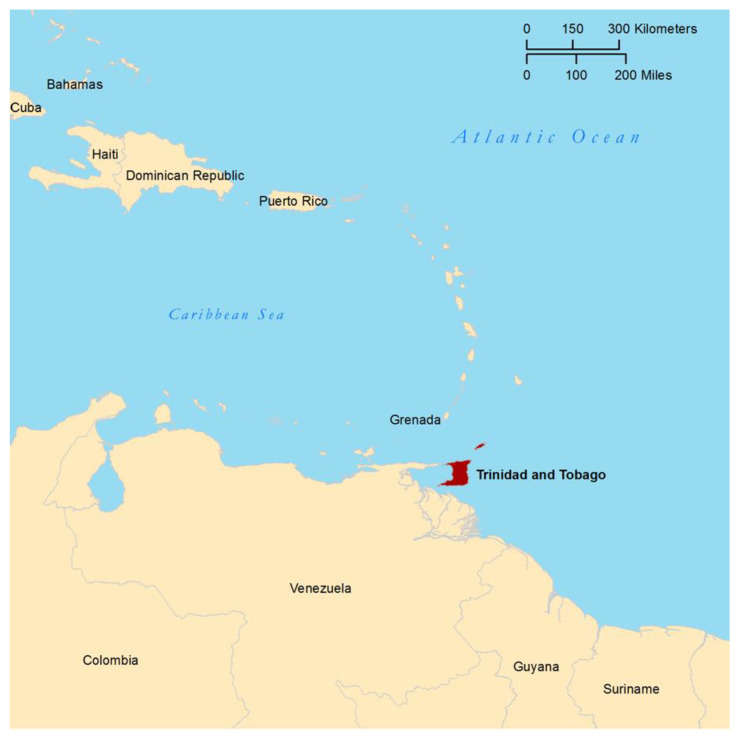
Location of Trinidad and Tobago relative to South America, Venezuela, and other Caribbean islands. Map created with the assistance of T. W. Shawa, Princeton Geographic Information Systems Librarian.

**Figure 2 tropicalmed-05-00166-f002:**
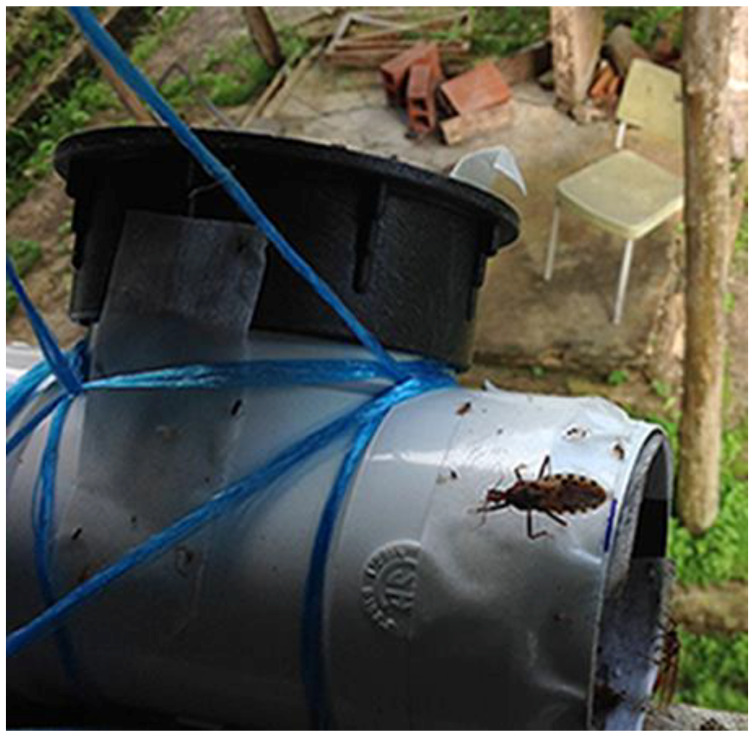
Successful mouse-baited trap. Adult *P. geniculatus* is caught on the adhesive material lining the outside of the trap. The trap is set on a windowsill, near an artificial light on the exterior of a ranger station in the village of Mt. Harris. Photo by J. K. Peterson.

**Figure 3 tropicalmed-05-00166-f003:**
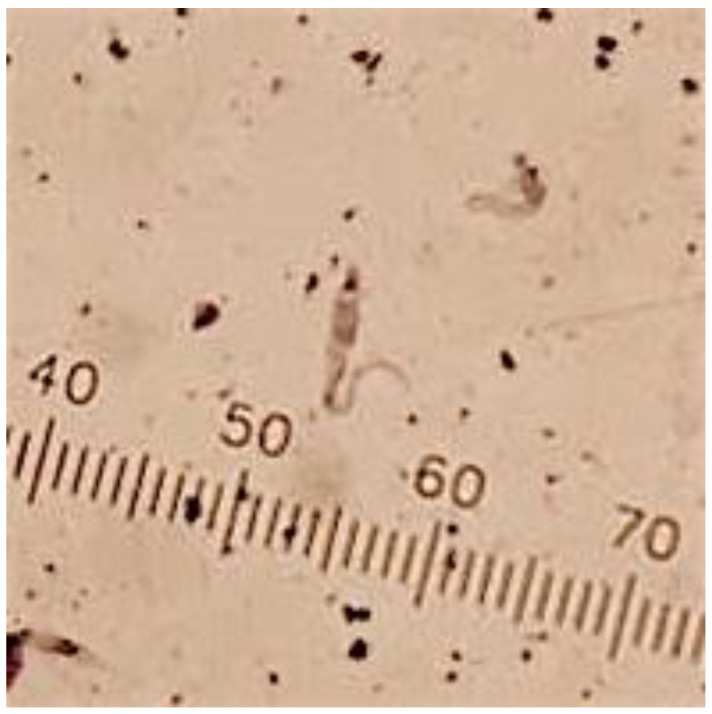
Giemsa stain of *T. cruzi* isolated from *P. geniculatus* collected in this study. Scale shown is in μm. Photo: J. K. Peterson.

**Figure 4 tropicalmed-05-00166-f004:**
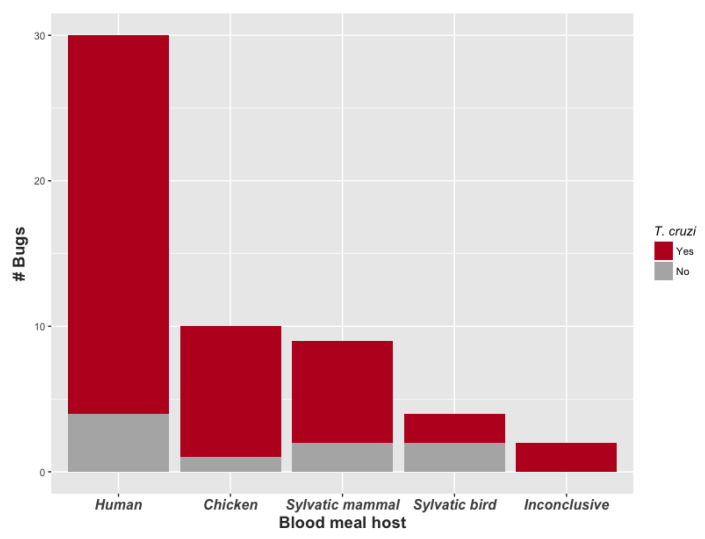
Number of bugs (i.e., blood meals) by blood meal host category and *T. cruzi* infection status of triatomine bug. Data were pooled across sites. One blood meal host was identified per bug. All triatomine species were adult *P. geniculatus* except for one adult male *R. pictipes*, which was infected with *T. cruzi*, and contained a human blood meal. Blood meal host species information is found in Table 2.

**Table 1 tropicalmed-05-00166-t001:** Triatomine Bug Collection Site Descriptions with *T. cruzi* Infection Status of Bugs Collected from Each Site, and Human Blood Meals Taken by the Bugs. Only Successful Collection Site Details Are Listed. Details for Every Sample Site Are in Appendix A.

Site Name	Location	Site Description	Bugs Collected ^1,2^			Human Blood Meals		
			Infected	Uninfected	Total	Infected ^3^	Uninfected ^3^	Total
Blanchisseuse	Northern coast	Coastal human home adjacent to secondary forest	1 (100%)	0	1	1 (100%)	0	1
Coal Mine	Central range ^4^	Scattered human homes adjacent to secondary forest	40 (87%)	6 (13%)	46	22 (92%)	2 (8%)	24
Matura	Northeast coast	Rural human home surrounded by secondary forest	1 (50%)	1 (50%)	2	0	1 (100%)	1
Mt Harris	Central range ^4^	Rural neighborhood in close proximity to secondary forest	3 (60%)	2 (40%)	5	2 (67%)	1 (33%)	3
Santa Cruz	Northern range ^4^	Laundry area of peri-urban human home	1 (100%)	0	1	1 (100%)	0	1

^1^ All bugs collected were *Panstrongylus geniculatus*, except for one *Rhodnius pictipes* male infected with *T. cruzi.* The *R. pictipes* individual was collected manually from the exterior wall of a house in Coal Mine. ^2^ Sites where just one bug was collected were public submissions, and not sites that were actively sampled. ^3^ Infection status refers to the bug with the blood meal, not the human.^4^ Central and northern ranges are the names of the east-west mountain ranges found in central and northern Trinidad.

**Table 2 tropicalmed-05-00166-t002:** Blood Meal Species Identified in Triatomine Bugs Collected, and *T. cruzi* Infection Status of the Bug or Bugs Corresponding to Each Species. All Data Are Available in Appendix A, Including Accession Numbers and Sequences.

Blood Meal Hosts ^1^			Triatomine Bugs ^2^		Collection Sites ^5^	BLAST Results ^3^	
Class	Genus/Species	Common Name	Blood Meals ^1^	*T. cruzi*-Positive Bugs ^4^		% Identity Match	E-Value
Mammalia	*Homo sapiens*	Human	30	26 (87%)	All	99%	1 × 10^−58^
	*Alouatta seniculus*	Red howler monkey	2	2 (100%)	CM, MH	98%	5 × 10^−134^
	*Dasyprocta leporina*	Red-rumped agouti	2	2 (100%)	CM, MT	100%	2 × 10^−128^
	*Coendou prehensilis*	Brazilian porcupine	1	1 (100%)	CM	100%	1 × 10^−144^
	*Herpestes javanicus*	Javan mongoose	1	1 (100%)	CM	100%	8 × 10^−146^
	*Metachirus nudicaudatus*	Brown four-eyed opossum	1	0 (0%)	CM	99%	8 × 10^−136^
	*Nectomys* ^6^	Water rat	1	1 (100%)	CM	98%	9 × 10^−138^
	*Proechimys trinitatis*	Trinidad spiny rat	1	0 (0%)	MH	100%	3 × 10^−64^
Aves	*Gallus gallus*	Chicken	10	9 (10%)	CM	99%	3 × 10^−106^
	*Pionus menstruus*	Blue-headed parrot	2	1 (50%)	CM	90%	2.5 × 10^−77^
	*Dryocopus lineatus*	Lineated woodpecker	1	1 (100%)	CM	99%	1 × 10^−92^
	*Pulsatrix perspicillata*	Spectacled owl	1	0 (0%)	CM	98%	0 *

^1^ One blood meal host was identified per bug. ^2^ All bugs were *P. geniculatus* except for one *R. pictipes* collected in Coal Mine, which was *T. cruzi* positive and contained a human blood meal. ^3^ Cut off for accepting an identity match in BLAST was ≥ 90%, and an e-value below 1 × 10^−10^. (The e-value describes the number of hits one can expect by chance, given database size.) Average % match and average e-value is shown for species found in more than one blood meal. ^4^ Positive *T. cruzi* infection in the bug does not mean positive host infection (and birds are not competent for *T. cruzi* infection). ^5^ CM = Coal Mine, MH = Mount Harris, MT = Matura. Site descriptions in Table 1. ^6^ BLAST hit corresponded to *Nectomys rattus*, which is not found in Trinidad. The cytb gene sequence for the congeneric species native to Trinidad (*Nectomys palmipes*) is not currently in the Genbank database. * E-values approaching 0 are rounded to 0 in BLAST.

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
