# Peer review of "Preliminary Characterization of Triatomine Bug Blood Meals on the Island of Trinidad Reveals Opportunistic Feeding Behavior on Both Human and Animal Hosts"

_tropicalmed, 2020, doi:10.3390/tropicalmed5040166_

Round 1
Reviewer 1 Report
Main concerns raised:
- Line 79 – “68% had detectable parasitemia (reference # 6)”. The number is to Hugh for detectable parasitemia in the supposed chronic phase of cardiomyopathy of Chagas disease, condition in which it is not easy to detect parasites in blood.
- Line 186 - Only one geographical area (northern and central area of the island) was selected to collection of triatomine bugs all but one of the same species. Also the majority of bugs were collected from a single rural area with scattered human dwellings located adjacent to secondary forest. Are the triatomine bugs used in the study representative of the whole Trinidad island.?
- The number of bugs collected can be considered small. What should be an statistical valid sample size?
- Line 204 and figure 4 - The cruzi contamination of bugs from humans is over 86%. If this fact represents the whole country the population is in risk of major complications by the infection.
- Line 297 on – The Authors agree that the sample is small, the geographic distribution and duration were limited. It was not clear if they intend to extend the study.
- Line 306 – “Finally, it must be kept in mind that a triatomine bug blood meal does not guarantee T. cruzi 307 transmission...” The cruzi transmission through triatomine bugs only happens if the bugs feed with the host blood. It is suggested the frase should be changed.
- Line 311 – The conclusions are inductive. They are not directly based on the results shown in the manuscript, as the sample is small and the findings are preliminary.
Reviewer 2 Report
The authors present the T. cruzi infection and blood meal sources in triatomine bugs (mainly Panstrongylus geniculatus) collected in domiciliary areas from Trinidad Island, Central America. Despite the low number of triatomines collected, limited geographic area and duration, the study took place in a very poor studied country, still considered non-endemic for Chagas disease. The presented results demonstrate high rates of T. cruzi infection in bugs collected in domiciliary areas, and high diversity of bugs’ blood meals, including almost 60% of humans. The study is well conducted, but I have some major conceptual concerns and other minor corrections.
MAJOR POINTS:
1) The One Health Approach, although very important, is mentioned in the title and abstract, but it does not appear anywhere else in the text, except in the last sentence of the manuscript. Moreover, the concept of One Health is much broader than it appears in this sentence. If the authors want to maintain the One Health Approach in the title and in the abstract, they should present this concept in the Introduction and properly discuss their findings in the light of this concept, pointing to possible solutions and approach for future studies. This would certainly improve the manuscript. Alternatively, authors may exclude that term and include something focused on the possibility of endemicity of the disease in Trinidad, which is a subject well addressed in the manuscript, especially in the discussion.
2) Introduction, lines 51-53: Review this sentence. Generalist species of triatomines are those that colonize different environments, such as wild and peridomiciliar and/or present distinct food sources. These insects are characterized by being opportunistic and, although capable of dispersing to other places, they only do this in adverse moments and in food deprivation. It is not characteristic of bugs to be frequently coming and going from different environments. The sentence on Discussion (lines 237-238) reinforces my point of view.
3) Introduction, lines 87-91: This statement is very simplistic concerning the distinct transmission routes to humans and needs to be revised. Transmission to humans is not dependent only on blood intake and defecation (vectorial contaminative route). The vectorial oral route has been extremely important in the North of South American continent (especially the Amazon region), due to accidental ingestion or food contaminated with feces from infected bugs. For this, insects can be attracted to homes by light and not only searching for food. And, mainly, the infection occurs regardless of whether or not this insect has human blood in its digestive tract.
4) Figure S1: A fundamental aspect that needs to be revised in the pamphlet is related to the possibility of oral infection. This transmission route was not cited, although it is currently the most important one in the North of South American continent. In protective measures, ways to prevent oral transmission must also be included.
5) Discussion, lines 226-227: I don’t agree with the connection stated. See above comment on the displacement of triatomines. The study demonstrates a generalist behavior of the main vector insect, feeding on different hosts, in addition to very high rates of infection by T. cruzi. I suggest authors to highlight this information.
6) Discussion, lines 254-256: Considering the T. cruzi infection in humans, this is important information that must be discussed. Contact with blood and carcasses of infected armadillos and opossums can potentially lead to human infection in the case of skin injury or oral accidental contamination, both for hunters and for buyers. Being so common in the region, it is a potential infection route that should be discussed.
7) Discussion, lines 309-310: Once more, the oral infection is ignored. The defecation time in no way influences vector competence when an infected bug is crushed with some food or its feces are accidentally ingested. This concept needs to appear more clearly in some parts of the document that were highlighted in my comments.
MINOR POINTS:
1) Figure 1: I consider this map unnecessary. It only reports the location of the country, easily found in a Google search for those who do not know.
2) Introduction, line 68: “…there are only few studies…”
3) Methodology, line 172: Inform characteristics of this PCR beads: name, manufacturer…
4) Methodology, line 176: The lowest percentage of identity described in table 2 is 90%. I suggest changing the criterion here from 85 to ≥ 90%. Also modify footnote 3 in table 2.
5) Results, lines 196-197: Identify the areas where the 6 positive insects in fresh stool examination were collected.
6) Table 2: “Birds”, instead of “Aves”
7) References: Revise in order to put all the scientific names in italics. This lack was noted for at least references 10, 15, 17, 33, 35, 36 and 37.
Reviewer 3 Report
This is a well written, well designed, and very interesting study of blood meals in triatomine bugs in Trinidad and their infection status with T. cruzi. I only have a few minor comments which might improve the paper:
State how the bugs were identified to species level in the M & Ms.
It would be useful if the authors included the sequences of the blood meals they obtained, at least in a supplementary file. Readers will be interested in the sizes of the amplification products obtained (relative to the expected +- 370) and the GenBank numbers of the BLAST best matches must be given.
Were any of the sequences from this study deposited in GenBank - it would be particularly useful to make the sequences available for the congeneric species - footnote 6,7 Table 2.
The M & Ms say the sequences were 'edited' - are Phred scores available?
It would add to the paper if more recent clinical data on Chagas disease in people on Trinidad could be provided; even if somewhat anecdotal as references from the 1980s and before are pretty old now? For example what do local cardiologists say?
Reviewer 4 Report
In the present manuscript, the authors have presented interesting evidence for a possible risk of infection for the population in Trinidad. This small study, which collected sample material over a short period of time, was able to show that a non-negligible proportion of triatomines are T. cruzi-positive. Using molecular biological methods, the host spectrum of the predatory bugs was determined to include humans as well as domestic and wild animals. This shows, as the authors emphasize, that Trinidad belongs to the 'Chagas-endemic' areas.
The study has of course its limitations due to the small sample size and the short observation period. But these are also clearly articulated by the authors. However, this study is a good basis for a larger study, which should cover different aspects, e.g.
- are there regional and seasonal variations in the occurrence of the bugs?
- where to find the larval stages and are these also infected with T. cruzi?
- what is the situation with other triatomines, are they also affected and also represent a potential danger for the human population?
- how do the bugs get infected, i.e. what are the (vertebrate) sources?
- what is the serum prevalence in human residents of regions where T.cruzi infected bugs are found?
- what is the serum prevalence of domestic and wild animals in the respective regions?
- are the mentioned heart diseases associated with Chagas?
The authors could briefly address these and similar questions in their summary as a lookout for a future large scale study. It is very important that the authors point out that this is not a purely enzootic phenomenon and that the One Health idea should play an important role here.
All in all, the study has been successful, only a few remarks could still be implemented by the authors:
- describe the selected sampling sites and their surroundings in more detail
- if data are available, briefly indicate whether it is known whether cases of Chagas' disease in humans are known at the respective locations
- add the description in Figure 2 that the bug was caught on the adhesive tape
- do not write "Blanchisseuse" and "Northern coast" bold in Table 1
- explain in Table 1 what the asterisk at "1*" means
- in Table 2 possibly change "Species" to "Species/Genus", because for Nectomys and Proechimys only the genus is known
- in Table 2 change "Pionus Menstruus" to "Pionus menstruus"
- in the introduction to the discussion, briefly discuss how the bugs could be infected (could humans be the biggest source here or is there possibly a reservoir in monkeys or possums?)
- please add commas in the first sentence of the discussion after "aimed" and after "charaterizing" to better understand the meaning of the sentence
Round 2
Reviewer 2 Report
Authors satisfactorily answered my questions and modified the manuscript following my request. The manuscript are ready to be published.
Reviewer 3 Report
I believe this paper can now be accepted.